# Knowledge of Sepsis in Nursing Students—A Cross-Sectional Study

**DOI:** 10.3390/ijerph182312443

**Published:** 2021-11-26

**Authors:** Gloria Valičević, Adriano Friganović, Biljana Kurtović, Cecilija Rotim, Sanja Ledinski Fičko, Sabina Krupa

**Affiliations:** 1Department of Anaesthesiology and Intensive Medicine, University Hospital Centre Zagreb, 10000 Zagreb, Croatia; adriano@hdmsarist.hr; 2Department of Nursing, University of Applied Health Sciences, 10000 Zagreb, Croatia; biljana.kurtovic@zvu.hr (B.K.); cecilijarotim@gmail.com (C.R.); sanja.ledinski-ficko@zvu.hr (S.L.F.); 3Andrija Štampar Teaching Institute of Public Health, 10000 Zagreb, Croatia; 4Institute of Health Sciences, College of Medical Sciences, University of Rzeszow, 35-310 Rzeszow, Poland; sabinakrupa@o2.pl

**Keywords:** sepsis, knowledge, nursing students

## Abstract

Background: Sepsis is defined as life-threatening organ dysfunction caused by an unregulated host response to infection. The emphasis is on the imbalance of homeostasis and the response to infection, as well as mortality and the importance of recognizing sepsis as early as possible. The knowledge of undergraduate nursing students is an extremely important indicator for future work in the healthcare system after graduation. The aim of this study was to investigate the levels of knowledge about sepsis among undergraduate nursing students and to compare differences in different years of study, as well as differences in their study model. Methods: A cross-sectional study was conducted on 618 nursing students at the University of Applied Health Sciences in Zagreb, Croatia. All three years of study and both full-time and part-time (employed) nursing students were included. The questionnaire “Determinants of Sepsis Knowledge” was used in the research. Results: The percentage and number of third-year students who correctly answered the items on Knowledge of Sepsis were statistically significant compared to the first two years of study. The percentage and number of employed students who responded correctly to the items on Knowledge of Sepsis were statistically significant compared to students who were not employed. Conclusions: The ability of nursing students to recognize and respond to the deterioration in a patient’s condition due to sepsis is very important, so appropriate education about sepsis is essential. We recommend a greater representation of sepsis content in the core curriculum of nursing students’ education in terms of theoretical instruction and clinical and simulation exercises.

## 1. Introduction

Sepsis was defined in 1992 as a systemic inflammatory response caused by infection [1]. According to the new definition from 2016, sepsis is defined as life-threatening organ dysfunction caused by an unregulated host response to infection. What is emphasized in the new definition is the imbalance of homeostasis and the response to the infection, as well as mortality and the importance of recognizing sepsis as early as possible [1]. 

Currently, sepsis and septic shock are global healthcare problems. The definition of sepsis was changed through the years. The first definition of sepsis was published in the 2000s. The guidelines for sepsis and septic shock were recommended for the treatment of sepsis. The first definition of sepsis was convened in Chicago in 1991. Following that, systemic inflammatory response syndrome, severe sepsis, sepsis, septic shock and multiple organ dysfunction syndromes began to be used in clinical practice. The definition of sepsis was: “two or more systemic inflammatory response syndrome criteria”, while severe sepsis was defined as clinical sepsis accompanied by organ dysfunction, hypotension or hypoperfusion. Septic shock is defined as a clinical tableau in which fluid/vasopressor-resistant hypotension (average artery blood pressure ≤70 mmHg) and hypoperfusion are observed. In 2001, the European Society of Intensive Care Medicine, the Society of Critical Care Medicine, the American Thoracic Society and the Surgical Infection Society held the second consensus and updated the criteria for sepsis. They changed signs and symptoms of sepsis, as early diagnosis is important to suspect infection-specific findings such as general, inflammatory, hemodynamic or organ dysfunction. The new definition of sepsis was: “a clinical syndrome combined with organ injury”, but they did not change old diagnostic criteria for sepsis. Severe sepsis was defined as “sepsis complicated by organ dysfunction”. A high level of disagreement with researchers and physicians led to mismatching. The diagnosis of sepsis and severe sepsis was similar and caused confusion when diagnosing. The newer definition Sepsis 3 defines sepsis as a “life threatening organ dysfunction caused by a dysregulated host response to infection”. Septic shock is defined as lactate levels rising above 2 mmol L^−1^ without hypovolemia and initiation of vasopressor treatment to keep mean arterial pressure above 65 mmHg. The Sequential Organ Failure Assessment (SOFA) is a scoring system for organ dysfunction. An increase of two or more in the SOFA scoring system indicates organ dysfunction. A report from the 2016 consensus claims that the use of the qSOFA in the intensive care unit helps detect the possibility of sepsis [2]. Research and data in Croatia about this issue have not been sufficiently solved. One study conducted in a university hospital on a sample of 314 cases in the period 2000–2005 indicated the percentage of admitted patients with sepsis increased from 3.7% (2000) to 11.7% (2005), while mortality increased from 14.4% (2000) to 20.3% (2005) [2]. Other significant data include the fact that, according to the Croatian Central Bureau of Statistics, sepsis is not listed among the 10 leading causes of death in Croatia; in the ranking of the deaths of elderly people in Croatia in 2013, it was in 12th place, which is probably because the diagnosis of sepsis is still insufficiently applied or recognized as the immediate cause of death [3]. Certain groups of people have an increased risk of developing sepsis: newborns and infants (under 1 year of age), the elderly (over 75 years of age) and people with a weakened immune system due to illness or addiction, including patients undergoing chemotherapy [4].

Various organizations were established to prevent sepsis and increase citizens’ awareness of it. The leading organization in the United States is the Global Sepsis Alliance, which has a vision of a world without sepsis and a mission to save lives and reduce suffering by raising awareness of sepsis as an emergency [5]. In 2012, they founded World Sepsis Day, which is celebrated every year on 13 September [6]. In Croatia, the Croatian Society of Nurses of Anaesthesia, Resuscitation, Intensive Care and Transfusion (HDMSARIST) launched a “Stop Sepsis” campaign, which educates medical staff and citizens about the symptoms, signs and prevention of sepsis [7]. 

A study by Poeze et al. investigated knowledge about definitions and pathophysiology of sepsis. The subjects were intensive care physicians and other specialist physicians caring for intensive care unit patients. Eighty-six percent of respondents agreed that the symptoms of sepsis could be misattributed to other conditions, and three-quarters of all interviewed physicians agreed that sepsis is a leading cause of mortality compared with other conditions in intensive care [8].

In their research, Harley et al. concluded that emergency nurses have a deficit in their knowledge of sepsis and the importance of a timely response in working with a patient with sepsis [9]. Their recommendations stated that the development of clinical guidelines for emergency room nurses would be useful [9]. Another study conducted by Storozuk et al., also with emergency nurses, confirmed the hypothesis of insufficient knowledge in the field of sepsis and the need for additional educational programs [10]. Maczuga and Kosson concluded in their study that nurses in emergency departments and intensive care units have a higher level of knowledge of how to recognize sepsis than other wards [9]. The authors concluded that it is extremely important to ensure the availability of appropriate high-quality information, as well as to raise awareness of the importance of knowledge of the early detection of sepsis [11].

In the study conducted by Harley et al. (2021), in which they assessed the knowledge of Australian nursing student’s in their final years of college, 50% of participants stated that they had heard of sepsis, while only 22% reported specific teaching lessons on sepsis, and only 44% of participants were able to identify the importance of sepsis early recognition. Most participants were able to correctly answer the question about sepsis symptom identification (86.1%). It is necessary to know the level of knowledge of nurses and conduct education so that nurses can provide quality and safe care to patients and influence the outcome of the patient’s condition. It is also important to identify the necessary standards for the safety and care of patients with sepsis and to introduce those standards into the nursing education curriculum [12]. 

Not many studies have been conducted on the topic of student knowledge; however, the knowledge of nurses in certain areas of the site has been researched. A similar study conducted at Tilton K. University of Florida called *Sepsis Knowledge in Undergraduate Nursing Students* was conducted on 40 respondents, of whom 77.5% were women, and 75% of respondents completed three semesters before participating in the study. The research identified the need for additional education of students about the characteristics of sepsis and screening measures for sepsis. There is a need to increase students’ knowledge of sepsis and a need to integrate sepsis into the curriculum of student studies [13]. Shime and co-workers conducted a survey on knowledge and perception of sepsis in Japan; the study involved doctors, nurses, medical students, nursing students and patients who came to the anesthesiology clinic, and a total of 588 respondents participated. Only one-quarter of students recognized the term sepsis, while those who recognized the term less than 40% knew the meaning of the term. As for the nurses themselves, one-third chose the correct definition of sepsis, related to knowledge of sepsis mortality; students believe that sepsis mortality is less than 10%, it could be related that such an opinion is associated with the incorrect perception of pathophysiology and severity of sepsis. The conclusion of the research is that additional education is required in the educational programs of medical staff and increased education of lay people about sepsis [13].

Although this is an important area, health professionals’ levels of knowledge have still not been a research subject in Croatia, where several studies were conducted on the prevalence of sepsis, but none on the level of knowledge. A review of the world literature found research conducted worldwide among nurses, but only a few studies on the student nurse population. According to Harley et al. internationally, it was recognized that nurses’ knowledge around sepsis is often limited [9]. The knowledge of undergraduate nursing students is an extremely important indicator for future work in the healthcare system after graduation and a topic of growing interest. The aim of this study was to investigate the levels of sepsis knowledge in the undergraduate nursing student population and to compare the differences across different years of study, as well as the differences in their study model. The importance of this study is in assessing the educational content that students receive in a particular year of study. 

## 2. Materials and Methods

### 2.1. Respondents and Procedure

A cross-sectional study was conducted at the University of Applied Health Sciences in Zagreb, Croatia, in 2019 and 2020. The University of Applied Health Sciences in Zagreb, Croatia, is the largest institution for the education of nurses, with the largest number of students from all over Croatia. The selected respondents were 618 undergraduate nursing students. All three years of study, and full-time and part-time (students employed in the healthcare system) students, were included. Participation in the research was voluntary; informed consent was obtained from all the subjects involved in the study. 

The lecturers at the university explained the goal and reason for the research to the students before the beginning of classes and distributed the questionnaires, which the students then completed.

### 2.2. Research Instruments

The instruments used in this study were demographic features (age, gender, year of study, employment) and the questionnaire designed by Eitze et al. [14]. For the purposes of the study, we carried out a linguistic validation, translation into Croatian and back-translation. The Kaiser–Meyer–Olkin measure of sampling adequacy gave a high value (KMO = 0.847, *p* < 0.001), indicating the suitability of factor analysis. Cronbach’s alpha was satisfactory at 0.76.

### 2.3. Data Analysis

The data analyses were carried out using IBM SPSS Statistics for Windows (version 22.0 [15]. IBM Corp: Armonk, NY, USA). Descriptive statistics (frequency, percentage, mean, standard deviation) were used to summarize the main characteristics of the sample.

The continuous variables (Knowledge and Symptoms) showed a statistically significant deviation from the Gauss normal distribution, but the descriptive parameters showed a median and interquartile range, which better represented the data. The normality of the distribution was verified using the Shapiro–Wilk test. For the nominal (categorical) variables, the number and percentage of participants are shown; the statistical significance between the variables was calculated using the Chi-square test. In order to test the statistical significance of the differences in the mean results on the Knowledge and Symptoms scale, the Mann–Whitney U (two independent samples) and the Kruskal–Wallis tests (more than two independent groups) were used.

## 3. Results

The respondents were 618 undergraduate nursing students, of which 504 (81.6%) were female, and 114 (18.4%) were male (Table 1).

The second-year students were found to have lower scores on the item “sepsis is a strong allergic reaction” than first- and third-year students. The second- and third-year students responded more accurately than first-year students to the item “sepsis has a strong bodily immune response”. The differences were also significant for the item “there are more cases of breast cancer than cases of sepsis”, where the third-year students best knew the correct answer, and for the item “sepsis can be caused by pneumonia”, the second- and third-year students had a higher number of correct answers than the first-year students (Table 2).

The employed students more often answered the item “sepsis is a strong allergic reaction” correctly. The same is true for the items “sepsis is caused by multidrug-resistant superbugs present in hospitals”, “sepsis can be diagnosed by a red line infiltrating from the wound to the heart”, “sepsis can be caused by pneumonia” and “sepsis can be caused by influenza” (Table 3).

The responses to the knowledge of sepsis symptoms by year of study were statistically significant in almost all items except for the penultimate one, in favor of better knowledge of second- and third-year students compared to first-years (Table 4).

The responses to the knowledge of sepsis symptoms according to whether the students were employed or not were statistically significant in the first five items, in favor of better knowledge of employed students (Table 5).

## 4. Discussion

Healthcare management expects graduate nurses to have the knowledge and skills appropriate to caring for patients with severe conditions such as sepsis. The nursing undergraduate studies’ core curriculum in Croatia includes the topic of sepsis in several courses, from pathogenesis and therapeutic procedures to nursing interventions. For example, it is included in the first year of courses such as Microbiology; in the second year of Nursing Care of Adults I, Infectiology and Internal Medicine; and in the third year of Nursing Care of Adults II [16].

In our study, we found that the undergraduate nursing students’ levels of knowledge about sepsis were insufficient, given the diversity and lack of accurate answers to questions about its etiology, pathogenesis and symptomatology. The results of our study are supported by the results of Harley et al., who found that final year graduate entry and undergraduate program students at five Australian universities had limited knowledge of sepsis (mean scores = 3.8/9; SD = 1.6) [12]. In their study, the education of students on sepsis was also conducted primarily through didactic lectures. We found that the levels of knowledge of sepsis in nursing students were higher in accordance with the enrolled higher year of study (Table 2 and Table 4). The results were expected to be higher, as students gained more theoretical knowledge and practical experience at clinical training sites during their years of study.

The results of our study determined that there are statistically significant differences in the levels of knowledge about sepsis in nursing students according to their study model (Table 3 and Table 5). We explain this by the fact that part-time students are employed, and in clinical practice, they have more opportunities to work with patients who may have sepsis. Moreover, part-time students, who are employed within the legally prescribed continual professional development in Croatia, have the opportunity to repeatedly educate themselves on the topic of sepsis. In a study conducted by Goulart et al., nurses ≥35 years of age had greater knowledge of the definition of sepsis (*p* = 0.042). Knowledge of high-volume fluid resuscitation (*p* = 0.001) and vasopressor application (*p* = 0.025) was higher in those with ≥10.5 years of professional experience [15]. The effect of continuing education on sepsis in employed nurses is supported by the findings of a study in which a multimodal design of one-year sepsis education of nurses was applied. In that study, the self-perceived frequency of use of competence behaviors was improved in the participants after the training, and the results of their knowledge after the test showed a significant improvement [17].

The importance of our study was also in the assessment of the educational content that students receive in a particular year of study. The representation of content on sepsis is insufficient because the topic of sepsis is limited in its theoretical content. The results of our study are a good basis for the content of sepsis to be better represented in theoretical and practical teaching units.

Clinical exercises provide students with the opportunity to directly interact with patient situational factors, in this case, sepsis, allowing them to practice clinical judgment and critical thinking [18]. Given the specificity of a condition such as sepsis, it is possible that during clinical exercises, a student may never encounter a patient with sepsis at all, so there is a need to develop simulation learning. The simulation presents a situation occurring in a clinical setting and allows the student to develop a perception of the possibilities that may arise when caring for patients and the unpredictable outcomes of clinical practice [19]. Such an educational strategy encourages the development of critical thinking and the ability to make clinical decisions, as well as the development of psychomotor skills in nursing students. In the development of this, some authors suggest the use of checklists in the assessment of knowledge and skills in simulation learning [20]. Diaz et al. found that the 21-item Early Detection of Paediatric Sepsis Assessment Checklist in a simulation using high-fidelity manikins was a valuable tool for assessing the students’ knowledge and their clinical readiness and performance through the safety and monitoring provided by simulation learning [21]. During the COVID-19 pandemic, live theoretical and practical classes have changed significantly, and teaching is largely conducted online worldwide. An example of good practice for interactive e-learning is the British e-learning package on the role of nurses in recognizing and implementing interventions in patients with sepsis. The package can be used by nursing students, newly hired nurses and other health professionals involved in the assessment and treatment of patients who may develop sepsis [22].

The limitation of this study is the lack of comparison with other higher education institutions in Croatia due to the lack of research conducted on this topic.

## 5. Conclusions

Sepsis is a leading cause of death and long-term complications in children and adults. Early recognition of its symptoms and therapeutic interventions can significantly contribute to an improvement in the patient’s condition. The ability of nursing students to recognize and respond to a patient’s deterioration due to sepsis is very important, so appropriate education about sepsis is essential. In our study, we investigated the levels of knowledge of undergraduate nursing students and found an overall lack of knowledge of students on recognizing, developing and managing sepsis, increasing knowledge of sepsis in accordance with a higher year of study, and a significant difference between full-time and part-time (employed) students. Given the small number of studies on this topic in the world, there is an urgent need for further research and the establishment of education that adequately prepares nursing students for prompt and appropriate care of patients with sepsis. The authors’ recommendation is a greater presence of content on sepsis in the core curriculum of nursing students’ education, in terms of theoretical classes and clinical and simulation exercises.

## Figures and Tables

**Table 1 ijerph-18-12443-t001:** Respondents’ demographic data.

		*n*	%
Gender	Female	504	81.6
	Male	114	18.4
Total		618	100
Age	17–19	71	11.5
	20	117	19
	21	92	14.9
	22–30	205	33.2
	31+	132	21.4
Total		617	100
Year of study	First year	404	65.4
	Second year	169	27.3
	Third year	45	7.3
Total		618	100
Employment	Yes	234	37.9
	No	383	62.1
Total		617	100
Study	Full-time	300	48.5
	Part-time	318	51.5
Total		618	100

**Table 2 ijerph-18-12443-t002:** Percentage and number of students with correct answers to “Knowledge of sepsis” according to the year of study.

		Year of Study						*p*
		First Year		Second Year		Third Year		
		*n*	%	*n*	%	*n*	%	
With sepsis, you must call the emergency services immediately.	Incorrect or unsure	85	21.0	37	21.9	5	11.1	0.259
	Correct	319	79.0	132	78.1	40	88.9	
Total		404	100	169	100	45	100	
Sepsis is an intense allergic reaction.	Incorrect or unsure	44	10.9	35	20.7	4	8.9	0.005
	Correct	360	89.1	134	79.3	41	91.1	
Total		404	100	169	100	45	100.0	
Sepsis is an intense immune response of the body.	Incorrect or unsure	128	31.7	29	17.2	7	15.6	<0.001
	Correct	276	68.3	140	82.8	38	84.4	
Total		404	100	169	100	45	100	
Sepsis is caused by multidrug-resistant superbugs in hospitals.	Incorrect or unsure	154	38.1	53	31.4	11	24.4	0.087
	Correct	250	61.9	116	68.6	34	75.6	
Total		404	100	169	100	45	100	
Sepsis can be diagnosed by a red line infiltrating from a wound up to the heart.	Incorrect or unsure	287	71.0	135	79.9	32	71.1	0.086
	Correct	117	29.0	34	20.1	13	28.9	
Total		404	100	169	100	45	100	
Mortality after heart attacks is higher than mortality after sepsis.	Incorrect or unsure	299	74	121	71.6	28	62.2	0.233
	Correct	105	26	48	28.4	17	37.8	
Total		404	100	169	100	45	100	
There are more cases of breast cancer than cases of sepsis.	Incorrect or unsure	304	75.2	116	68.6	27	60.0	0.043
	Correct	100	24.8	53	31.4	18	40	
Total		404	100	169	100	45	100	
Sepsis can be caused by lung inflammation.	Incorrect or unsure	187	46.3	48	28.4	11	24.4	<0.001
	Correct	217	53.7	121	71.6	34	75.6	
Total		404	100	169	100	45	100	
Sepsis can be caused by influenza.	Incorrect or unsure	261	64.6	92	54.4	28	62.2	0.074
	Correct	143	35.4	77	45.6	17	37.8	
Total		404	100	169	100	45	100	

**Table 3 ijerph-18-12443-t003:** Percentage and number of students with correct answers to “Knowledge of sepsis” according to employment.

		Employment				*p*
		Yes		No		
		*n*	%	*n*	%	
With sepsis, you must call the emergency services immediately.	Incorrect or unsure	58	24.8	68	17.8	0.040
	Correct	176	75.2	315	82.2	
Total		234	100.0	383	100	
Sepsis is an intense allergic reaction.	Incorrect or unsure	16	6.8	66	17.2	<0.001
	Correct	218	93.2	317	82.8	
Total		234	100	383	100	
Sepsis is an intense immune response of the body.	Incorrect or unsure	60	25.6	103	26.9	0.778
	Correct	174	74.4	280	73.1	
Total		234	100	383	100	
Sepsis is caused by multidrug-resistant superbugs in hospitals.	Incorrect or unsure	66	28.2	151	39.4	0.005
	Correct	168	71.8	232	60.6	
Total		234	100	383	100	
Sepsis can be diagnosed by a red line infiltrating from a wound up to the heart.	Incorrect or unsure	137	58.5	316	82.5	<0.001
	Correct	97	41.5	67	17.5	
Total		234	100.0	383	100	
Mortality after heart attacks is higher than mortality after sepsis.	Incorrect or unsure	169	72.2	278	72.6	0.926
	Correct	65	27.8	105	27.4	
Total		234	100	383	100	
There are more cases of breast cancer than cases of sepsis.	Incorrect or unsure	165	70.5	281	73.4	0.459
	Correct	69	29.5	102	26.6	
Total		234	100	383	100	
Sepsis can be caused by lung inflammation.	Incorrect or unsure	51	21.8	194	50.7	<0.001
	Correct	183	78.2	189	49.3	
Total		234	100	383	100	
Sepsis can be caused by influenza.	Incorrect or unsure	106	45.3	274	71.5	<0.001
	Correct	128	54.7	109	28.5	
Total		234	100	383	100	

**Table 4 ijerph-18-12443-t004:** Percentage and number of students with correct answers to “Knowledge of the symptoms of sepsis” according to their year of study.

		Year of Study						*p*
		First Year		Second Year		Third Year		
		*n*	%	*n*	%	*n*	%	
Are chills and fever symptoms of sepsis?	Incorrect or unsure	135	33.4	32	18.9	6	13.3	<0.001
	Correct	269	66.6	137	81.1	39	86.7	
Total		404	100	169	100	45	100	
Is disorientation a symptom of sepsis?	Incorrect or unsure	164	40.6	57	33.7	6	13.3	0.001
	Correct	240	59.4	112	66.3	39	86.7	
Total		404	100	169	100	45	100	
Is shortness of breath a symptom of sepsis?	Incorrect or unsure	285	70.5	100	59.2	20	44.4	<0.001
	Correct	119	29.5	69	40.8	25	55.6	
Total		404	100	169	100	45	100	
Is a high heart rate a symptom of sepsis?	Incorrect or unsure	146	36.1	40	23.7	8	17.8	0.002
	Correct	258	63.9	129	76.3	37	82.2	
Total		404	100	169	100	45	100	
Is low blood pressure a symptom of sepsis?	Incorrect or unsure	251	62.1	84	49.7	18	40	0.001
	Correct	153	37.9	85	50.3	27	60.0	
Total		404	100	169	100	45	100	
Is diarrhoea a symptom of sepsis?	Incorrect or unsure	218	54.0	98	58.0	18	40	0.099
	Correct	186	46.0	71	42.0	27	60.0	
Total		404	100	169	100	45	100	
Are a skin rash and eczema symptoms of sepsis?	Incorrect or unsure	272	67.3	89	52.7	26	57.8	0.003
	Correct	132	32.7	80	47.3	19	42.2	
Total		404	100	169	100	45	100	

**Table 5 ijerph-18-12443-t005:** Percentage and number of students with correct answers to “Knowledge of the symptoms of sepsis” according to employment.

	Employment	*p*
Yes		No	
*n*	%	*n*	%
Are chills and fever symptoms of sepsis?	Incorrect or unsure	22	9.4	150	39.2	<0.001
Correct	212	90.6	233	60.8	
Total		234	100	383	100	
Is disorientation a symptom of sepsis?	Incorrect or unsure	48	20.5	178	46.5	<0.001
Correct	186	79.5	205	53.5	
Total		234	100	383	100	
Is shortness of breath a symptom of sepsis?	Incorrect or unsure	117	50	287	74.9	<0.001
Correct	117	50	96	25.1	
Total		234	100	383	100	
Is a high heart rate a symptom of sepsis?	Incorrect or unsure	46	19.7	147	38.4	<0.001
Correct	188	80.3	236	61.6	
Total		234	100	383	100	
Is low blood pressure a symptom of sepsis?	Incorrect or unsure	102	43.6	250	65.3	<0.001
Correct	132	56.4	133	34.7	
Total		234	10	383	100	
Is diarrhoea a symptom of sepsis?	Incorrect or unsure	115	49.1	218	56.9	0.067
Correct	119	50.9	165	43.1	
Total		234	100	383	100	
Are a skin rash and eczema symptoms of sepsis?	Incorrect or unsure	147	62.8	239	62.4	0.932
Correct	87	37.2	144	37.6	
Total		234	100	383	100

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
