# Peer review of "Knowledge of Sepsis in Nursing Students—A Cross-Sectional Study"

_ijerph, 2021, doi:10.3390/ijerph182312443_

Round 1

Reviewer 1 Report

Thank you for the opportunity to review this manuscript. This study is to investigate the levels of knowledge about sepsis among undergraduate nursing students. After carefully review this paper, the following comments are listed for your reference:

The general population knowledge and knowledge among healthcare students are two significantly different themes. I suggest that introduction focuses on healthcare professionals including students rather than general population. But there is a lot about the findings in public awareness of sepsis in Germany or USA, which is not the focus of your research. The focus should be professional people’s cognition and knowledge of sepsis, but the authors provide few reference to existing work in this area.

The authors mentioned that “the authors’ recommendation is a greater presence of content on sepsis in the core curriculum of nursing students' education, in terms of theoretical classes and clinical and simulation exercises”. Without clarifying the reasons for the cognitive differences between students in different grades, such suggestions are slightly subjective.

They did not adequately improve their research design. They did not improve the representativeness of samples. Thus, I don’t know why they resubmit this manuscript without any necessary improvement. 

Author Response

A: Your suggestion was followed/applied at page 2,line 69-87 we put more text about knowledge of students, and page 3, line 101-110.

Line 69-87: '' Not much research has been conducted on the topic of student knowledge research, what has been researched is the knowledge of nurses in certain areas of the site. A similar study conducted at Tilton K. University of Florida called Sepsis Knowledge in Undergraduate Nursing Students was conducted on 40 respondents of whom 77.5% were women, and 75% of respondents completed 3 semesters before participating in the study. The research identified the need for additional education of students about the characteristics of sepsis and screening measures for sepsis. There is a need to increase students' knowledge of sepsis, a need to integrate sepsis into the curriculum of student studies (6). Shime and co-workers conducted a survey on knowledge and perception of sepsis in Japan, the study involved doctors, nurses, medical students, nursing students and patients who came to the anesthesiology clinic, a total of 588 respondents participated. Only one quarter of students recognized the term sepsis, while those who recognized the term less than 40% knew the meaning of the term. As for the nurses themselves, one third chose the correct definition of sepsis, related to knowledge of sepsis mortality, students believe that sepsis mortality is less than 10%, it can be related that such an opinion is associated with incorrect perception of pathophysiology and severity of sepsis. The conclusion of the research is that additional education is needed in the educational programs of medical staff and increased education of lay people about sepsis (7).''

In the study conducted by Harley et al (2021), in which they assessed the knowledge of Australian nursing student's in their final years of college, 50% of participants stated that they had heard of sepsis, while only 22% reported specific teaching lessons on sepsis, and only 44% of participants were able to identify the importance of sepsis early recognition. Most participants were able to correctly answer the question about sepsis symptom identification (86.1%). It is necessary to know the level of knowledge of nurses and conduct education so that nurses can provide quality and safe care to patients and influence the outcome of the patient's condition. It is also important to identify the necessary standards for the safety and care of patients with sepsis and to introduce those standards into the nursing education curriculum [12].

A: Your suggestion was followed/applied at page 2, line 90-112, text about knowledge of general population was removed.

Reviewer 2 Report

I am very glad that this manuscript has been revised accordingly.

Author Response

A: Your suggestion was followed/applied at page 2,line 69-87 we put more text about knowledge of students, and page 3, line 101-110.

Line 69-87: '' Not much research has been conducted on the topic of student knowledge research, what has been researched is the knowledge of nurses in certain areas of the site. A similar study conducted at Tilton K. University of Florida called Sepsis Knowledge in Undergraduate Nursing Students was conducted on 40 respondents of whom 77.5% were women, and 75% of respondents completed 3 semesters before participating in the study. The research identified the need for additional education of students about the characteristics of sepsis and screening measures for sepsis. There is a need to increase students' knowledge of sepsis, a need to integrate sepsis into the curriculum of student studies (6). Shime and co-workers conducted a survey on knowledge and perception of sepsis in Japan, the study involved doctors, nurses, medical students, nursing students and patients who came to the anesthesiology clinic, a total of 588 respondents participated. Only one quarter of students recognized the term sepsis, while those who recognized the term less than 40% knew the meaning of the term. As for the nurses themselves, one third chose the correct definition of sepsis, related to knowledge of sepsis mortality, students believe that sepsis mortality is less than 10%, it can be related that such an opinion is associated with incorrect perception of pathophysiology and severity of sepsis. The conclusion of the research is that additional education is needed in the educational programs of medical staff and increased education of lay people about sepsis (7).''

In the study conducted by Harley et al (2021), in which they assessed the knowledge of Australian nursing student's in their final years of college, 50% of participants stated that they had heard of sepsis, while only 22% reported specific teaching lessons on sepsis, and only 44% of participants were able to identify the importance of sepsis early recognition. Most participants were able to correctly answer the question about sepsis symptom identification (86.1%). It is necessary to know the level of knowledge of nurses and conduct education so that nurses can provide quality and safe care to patients and influence the outcome of the patient's condition. It is also important to identify the necessary standards for the safety and care of patients with sepsis and to introduce those standards into the nursing education curriculum [12].

A: Your suggestion was followed/applied at page 2, line 90-112, text about knowledge of general population was removed.

This manuscript is a resubmission of an earlier submission. The following is a list of the peer review reports and author responses from that submission.

Round 1

Reviewer 1 Report

In each country the mortality from wrong diagnosis of sepsis is quite high and therefore it is important to recognize how well our healthcare professionals are trained to diagnose the common signs and symptoms of sepsis in different settings and different age groups.

There is no doubt that research among healthcare professionals about knowledge of sepsis is not well published. For that reason I think that this study is very important for the field.

Suggestions for improvement:

1. The aim of the study presented here was to find the level of knowledge about sepsis among nursing students in Croatia. Authors referred to some important sources of information in their references, however I would expect a more comprehensive approach to literature review and including other international papers such as for example M.M. Levy, et al.(2003), particularly important for the distinguishing between so called severe sepsis and  septic shock. There are resources that refer to the important evolution of definition of sepsis, changes to management and criteria that are used in healthcare professionals’ undergraduate training. I do not feel that this has been sufficiently covered and explained in introduction. This is important from the point of view of international nursing/ medical curricula as well as leading to studies on students' level of knowledge about sepsis.

2. I am slightly confused what the authors trying to say in introduction. Are they refer to the general population knowledge or knowledge among healthcare students? These are two significantly different themes. I suggest that introduction focuses on healthcare professionals including students rather than general population. This will provide a nice lead the studies performed by the authors.

3. The questionnaire used is appropriate and based on published literature. The analysis of the results is clear for a reader. The presentation of the results in the table 2, however, could be better formatted. Within the document I have,  it is difficult to notice “Total” as well as making sense of particular responses.

Overall, this is a comprehensive approach to determine the level of knowledge of sepsis among undergraduate nursing students. This study will be helpful in future to determine whether healthcare professionals’ undergraduate curricula in other countries need improvement in this area. This is important as healthcare professionals move between countries and often need to take additional exams and qualifications to reach particular standards. It would be interesting to compare the level of knowledge on the subject among different countries.

Author Response

REVIEWER #1

  1. The aim of the study presented here was to find the level of knowledge about sepsis among nursing students in Croatia. Authors referred to some important sources of information in their references, however I would expect a more comprehensive approach to literature review and including other international papers such as for example M.M. Levy, et al.(2003), particularly important for the distinguishing between so called severe sepsis and  septic shock. There are resources that refer to the important evolution of definition of sepsis, changes to management and criteria that are used in healthcare professionals’ undergraduate training. I do not feel that this has been sufficiently covered and explained in introduction. This is important from the point of view of international nursing/ medical curricula as well as leading to studies on students' level of knowledge about sepsis.

A: Your suggestion was followed/applied at page 1 to 2. We added text about development diagnosis of sepsis and severe sepsis.

Line: 38-64: ''Today, sepsis and septic shock are global healthcare problems. Definition of sepsis was changed through the years. First definition of sepsis was published in the 2000s. In the guidelines for sepsis and septic shock was recommend for treatment sepsis. Fist definition of sepsis was convened in Chicago in 1991. After that systemic inflammatory response syndrome, severe sepsis, sepsis, septic shock and multiple organ dysfunction syndrome began to be used in clinical practice. Definition of sepsis was: ‘’two or more systemic inflammatory response syndrome criteria, while severe sepsis was defined as clinical sepsis accompanied by organ dysfunction, hypotension or hypoperfusion. Septic shock is defined as a clinical tableau in which fluid/vasopressor-resistant hypotension (average artery blood pressure ≤70 mmHg) and hypoperfusion is observed. After that, in 2001, the European Society of Intensive Care Medicine, the Society of Critical Care Medicine and American Thoracic Society and the Surgical Infection Society held the second consensus and updated the criteria for sepsis. They changed signs and symptoms of sepsis, for the early diagnosis is important to suspect about infection-specific findings like general, inflammatory, hemodynamic or organ dysfunction. New definition of sepsis was: ‘’a clinical syndrome combined with organ injury, but they did not change old diagnostic criteria for sepsis. Severe sepsis was defined as ''sepsis complicated by organ dysfunction''. Because of high-level of disagreement with researchers and physicians led to mismatching. Diagnosis of sepsis and severe sepsis was similar and made a confusion of sense for diagnose sepsis or severe sepsis. Newer definition Sepsis 3 defines sepsis as a ‘life.threatening organ dysfunction caused by a dysregulated host response to infection’. Septic shock is defined as lactate levels rising above 2 mmol L-1 without hypovolemia and initiation of vasopressor treatment to keep mean arterial pressure above 65 mmHg. The Sequential Organ Failure Assessment (SOFA) is scoring system for organ dysfunction. Increase of two or more SOFA scoring system meaning of organ dysfunction. Report from the 2016 consensus claims that use of the qSOFA in the intensive care unit helps detect the possibility of sepsis.

Line: 88-92: Poeze et al. in their research about knowledge about definitions and pathophysiology of sepsis. Subject were intensive care physicians and other specialist physicians caring for intensive care unit patients. 86% respondent agreed that the symptoms of sepsis can be misattributed to other conditions, three-quarters of all interviewed physicians agreed that sepsis is leading cause of mortality compared with other conditions in intensive care.

  1. I am slightly confused what the authors trying to say in introduction. Are they refer to the general population knowledge or knowledge among healthcare students? These are two significantly different themes. I suggest that introduction focuses on healthcare professionals including students rather than general population. This will provide a nice lead the studies performed by the authors.

A: Your suggestion was followed/applied at page 2, line 58-83, and text about knowledge of general population was removed.

A: Your suggestion was followed/applied at page 2,line 69-87 we put more text about knowledge of students.

Line 69-87: '' Not much research has been conducted on the topic of student knowledge research, what has been researched is the knowledge of nurses in certain areas of the site. A similar study conducted at Tilton K. University of Florida called Sepsis Knowledge in Undergraduate Nursing Students was conducted on 40 respondents of whom 77.5% were women, and 75% of respondents completed 3 semesters before participating in the study. The research identified the need for additional education of students about the characteristics of sepsis and screening measures for sepsis. There is a need to increase students' knowledge of sepsis, a need to integrate sepsis into the curriculum of student studies (6). Shime and co-workers conducted a survey on knowledge and perception of sepsis in Japan, the study involved doctors, nurses, medical students, nursing students and patients who came to the anesthesiology clinic, a total of 588 respondents participated. Only one quarter of students recognized the term sepsis, while those who recognized the term less than 40% knew the meaning of the term. As for the nurses themselves, one third chose the correct definition of sepsis, related to knowledge of sepsis mortality, students believe that sepsis mortality is less than 10%, it can be related that such an opinion is associated with incorrect perception of pathophysiology and severity of sepsis. The conclusion of the research is that additional education is needed in the educational programs of medical staff and increased education of lay people about sepsis (7).''

  1. The questionnaire used is appropriate and based on published literature. The analysis of the results is clear for a reader. The presentation of the results in the table 2, however, could be better formatted. Within the document I have,  it is difficult to notice “Total” as well as making sense of particular responses.

A: Your suggestion was followed/applied at page 4,line 144-145. ''Total'' moved to be more noticeable.

Reviewer 2 Report

Thank you for sending your paper entitled “How can we prevent sepsis? Knowledge of sepsis in nursing students-a cross-sectional study.” to International Journal of Environmental Research and Public Health. After carefully review this interesting paper, the following comments are listed for your reference:

    1. To ensure that the manuscript is able to advance to publication status, authors should ensure a thorough English and grammatical edit across the full manuscript. If sentences are not constructed in a grammatically correct manner it can impact meaning in unintended ways.To increase potential citations, authors should check keywords against those recommended in the MESH Browser of Medical Subject Headings https://meshb.nlm.nih.gov/search
    2. Abstract (P1, L18-21): It would be desirable to include when the study was carried out in the abstract section.
    3. Methods (P3, L110-111): What do you mean by “full-time and part-time employed students”? How does it work? For international readers, I would recommend a more detailed explanation.
    4. Results (P3-9): What these findings add to the field? Third-year students are expected to have a higher level of knowledge than first- and second-year students. You compare levels of knowledge between years and study models, but there is not information about whether or not the levels of knowledge that students have are insufficient. On the other hand, what would it be based on if the levels of knowledge are sufficient or insufficient? Is there any kind of instruction on the questionnaire about how many correct items are optimal for having good or bad sepsis knowledge?
    5. Discussion (P8-9): I would suggest improving the discussion section. If it was difficult due to lack of literature, this should be mentioned in the limitations, along with other study limitations.
    6. Limitations (P10, L228): I would recommend including a limitations section at the end of the discussion.
    7. References (P10-11): The references style need to be checked.

Author Response

REVIEWER #2

1.To ensure that the manuscript is able to advance to publication status, authors should ensure a thorough English and grammatical edit across the full manuscript. If sentences are not constructed in a grammatically correct manner it can impact meaning in unintended ways.To increase potential citations, authors should check keywords against those recommended in the MESH Browser of Medical Subject Headings https://meshb.nlm.nih.gov/search

A: Manuscript has been corrected by native speaker proof-reader and certificate for proofreading is attached to the supporting documents-

  1. Abstract (P1, L18-21): It would be desirable to include when the study was carried out in the abstract section.

A: Your suggestion was followed/applied at page 1, line 18-21, we added time when the study was carried out.

Line:18-20: ‘’A cross-sectional study was conducted on 618 nursing students at the University of Applied Health Sciences in Zagreb, Croatia in 2019 and 2020.’’

3.Methods (P3, L110-111): What do you mean by “full-time and part-time employed students”? How does it work? For international readers, I would recommend a more detailed explanation.

A: Your suggestion was followed/applied at page 4, as follow: part-time students (students employed in healthcare system)

4.Results (P3-9): What these findings add to the field? Third-year students are expected to have a higher level of knowledge than first- and second-year students. You compare levels of knowledge between years and study models, but there is not information about whether or not the levels of knowledge that students have are insufficient. On the other hand, what would it be based on if the levels of knowledge are sufficient or insufficient? Is there any kind of instruction on the questionnaire about how many correct items are optimal for having good or bad sepsis knowledge?

A: The assessment of students' knowledge was performed using the statistical methods listed in the Results section.

5.Discussion (P8-9): I would suggest improving the discussion section. If it was difficult due to lack of literature, this should be mentioned in the limitations, along with other study limitations.

A: Your suggestion was followed/applied at page 11, as follow: The limitation of this study is the lack of comparison with other higher education institutions in Croatia due to the lack of research conducted on this topic.

  1. Limitations (P10, L228): I would recommend including a limitations section at the end of the discussion.

Your suggestion was followed/applied at page 10, we added limitations as we mentioned in the answer in comment No5.

  1. References (P10-11): The references style need to be checked

Your suggestion was followed/applied at page 10-11. The references has been checked and corrected.

Reviewer 3 Report

Thank you for the opportunity to review this manuscript. While I believe the topic (as suggested by the title) is an important one, I am not sure if the author has adequately/accurately addressed the study aims according to the study title.

In abstract, “the ability of nursing students to recognize and respond to the deterioration in a patient’s condition due to sepsis is very important, so appropriate education about sepsis is essential. We recommend a greater representation of sepsis content in the core curriculum of nursing students' education in terms of theoretical instruction, clinical and simulation exercises.” It looks like the background rather than conclusion. The conclusions should be supported by the results.

The authors mentioned “the biggest problem is the lack of knowledge and information, but low public awareness of sepsis and the late recognition and treatment of sepsis are also very concerning, and has led to an annual increase in cases of 8-13% over the last decade” (Lines 75-77). What area was this survey results obtained from?

In literature review, there is a lot in there (Lines 58-83) about the findings in public awareness of sepsis in Germany, USA and Ireland, which is not the focus of your research. The focus should be the (emergency) nurses’ cognition and knowledge of sepsis, but the authors provide few reference to existing work in this area.

As to research question, the authors mentioned “although this is an important area, health professionals’ levels of knowledge have still not been a research subject in Croatia, where several studies have been conducted on the prevalence of sepsis, but none on the level of knowledge.” Is it the real situation, or is it because the authors did not give a comprehensive review? Authors should clearly state the research problem or knowledge gap (No previous studies address this issue? Or this issue has not been sufficiently solved? A clear statement of research problem should be provided). The authors also mentioned “a review of the world literature found research conducted worldwide among nurses, but only a few studies on the student nurse population”. Please provide evidence or a reference of this review. In addition, it is interesting to consider why few studies on the student nurse population or why there has been no concern on this topic? This point forks into at least two issues: they don't think that there is any question/confusion on sepsis knowledge improvement of undergraduate nursing students; or they don't care about the students' mastery of knowledge when they are in universities, although “the knowledge of undergraduate nursing students is an extremely important indicator for future work in the healthcare system after graduation”. 

In data collection, I don’t understand why the convenience sample was adopted, because this research seems to focus on a local issue of Croatia. What exactly is the contribution of conducting an assessment of the educational content that students receive in a particular year of study? What’s more, please provide the reason for selecting students in University of Applied Health Sciences in Zagreb. It seems to be a mismatch between the sampling of respondents and the aim of the study.

The sampling is unequal/unbalanced from the point of “Year of study” -First year (65.4%), Second year (27.3%), and Third year (7.3%). Certainly, the results of this research require confirmation in a wider sample, for generalisability purposes. 

In questionnaire survey, the detail in how the survey instrument was developed over an iterative process should be provided. High-level transparency is critical, and it can give confidence in the authors' processes. I can’t find the question design on “Knowledge of sepsis” or “Knowing the symptoms of sepsis”. The attributes should not be subjectively provided by the author(s) as is. If there is no theoretical foundation of the attribute framework being proposed, experts should be invited to confirm the appropriateness of the framework.

The English language needs significant improvement. I would suggest getting an English language editor to proofread prior to next round of submission.

The whole research process is not rigorous, and the novelty is marginal.

Author Response

REVIEWER #3

  1. In abstract, “the ability of nursing students to recognize and respond to the deterioration in a patient’s condition due to sepsis is very important, so appropriate education about sepsis is essential. We recommend a greater representation of sepsis content in the core curriculum of nursing students' education in terms of theoretical instruction, clinical and simulation exercises.” It looks like the background rather than conclusion. The conclusions should be supported by the results.

A: Your suggestion was followed/applied, as follow: The aim of this study was to investigate the levels of knowledge about sepsis amongof undergraduate nursing students and to compare differences in different years of study, as well as differences in their study model.

  1. The authors mentioned “the biggest problem is the lack of knowledge and information, but low public awareness of sepsis and the late recognition and treatment of sepsis are also very concerning, and has led to an annual increase in cases of 8-13% over the last decade” (Lines 75-77). What area was this survey results obtained from?

A:Due to comments of the Reviewer 1, this part is deleted.

  1. In literature review, there is a lot in there (Lines 58-83) about the findings in public awareness of sepsis in Germany, USA and Ireland, which is not the focus of your research. The focus should be the (emergency) nurses’ cognition and knowledge of sepsis, but the authors provide few reference to existing work in this area.

A: Due to comments of the Reviewer 1, this part is changed.

  1. As to research question, the authors mentioned “although this is an important area, health professionals’ levels of knowledge have still not been a research subject in Croatia, where several studies have been conducted on the prevalence of sepsis, but none on the level of knowledge.” Is it the real situation, or is it because the authors did not give a comprehensive review? Authors should clearly state the research problem or knowledge gap (No previous studies address this issue? Or this issue has not been sufficiently solved? A clear statement of research problem should be provided). The authors also mentioned “a review of the world literature found research conducted worldwide among nurses, but only a few studies on the student nurse population”. Please provide evidence or a reference of this review. In addition, it is interesting to consider why few studies on the student nurse population or why there has been no concern on this topic? This point forks into at least two issues: they don't think that there is any question/confusion on sepsis knowledge improvement of undergraduate nursing students; or they don't care about the students' mastery of knowledge when they are in universities, although “the knowledge of undergraduate nursing students is an extremely important indicator for future work in the healthcare system after graduation”. 

A: Your suggestion was followed/applied, as follow:

 Research and data in Croatia about this issue has not been sufficiently solved

According to Harley et al. internationally, it has been recognised that nurse’s knowledge around sepsis is often limited [11]. so Tthe knowledge of undergraduate nursing students is an extremely important indicator for future work in the healthcare system after graduation and topic of growing interest.

  1. In data collection, I don’t understand why the convenience sample was adopted, because this research seems to focus on a local issue of Croatia. What exactly is the contribution of conducting an assessment of the educational content that students receive in a particular year of study? What’s more, please provide the reason for selecting students in University of Applied Health Sciences in Zagreb. It seems to be a mismatch between the sampling of respondents and the aim of the study.

A: Your suggestion was followed/applied in Results Section, as follow: University of Applied Health Sciences in Zagreb, Croatia is the largest institution for the education of nurses with the largest number of students from all over Croatia.

  1. The sampling is unequal/unbalanced from the point of “Year of study” -First year (65.4%), Second year (27.3%), and Third year (7.3%). Certainly, the results of this research require confirmation in a wider sample, for generalisability purposes. 

A: We appreciate your advice, which we'll follow in the future studies about sepsis knowledge.

  1. In questionnaire survey, the detail in how the survey instrument was developed over an iterative process should be provided. High-level transparency is critical, and it can give confidence in the authors' processes. I can’t find the question design on “Knowledge of sepsis” or “Knowing the symptoms of sepsis”. The attributes should not be subjectively provided by the author(s) as is. If there is no theoretical foundation of the attribute framework being proposed, experts should be invited to confirm the appropriateness of the framework.

A: We appreciate your comment, on the Page 4 we have explained the development and translation of the instrument, as follow: The instruments used in this study were demographic features (age, gender, year of study, employment), and the questionnaire designed by Eitze et al [7]. For the purposes of the study we carried out a linguistic validation, translation into Croatian, and back translation.

  1. The English language needs significant improvement. I would suggest getting an English language editor to proofread prior to next round of submission.

A: Proofreading certificate has been attached in the manuscript submission process.

  1. The whole research process is not rigorous, and the novelty is marginal.

A: We appreciate your comment, we’ll take all of the abovementioned concerns in our future work.

Round 2

Reviewer 1 Report

I can see that authors have now included more relevant literature.

I feel that there is an issue with English and grammar. There is an overuse of "research" word and it makes no grammatical sense ("Line 69-87: '' Not much research has been conducted on the topic of student knowledge research, what has been researched is the knowledge of nurses in certain areas of the site).

I would like to advise a comprehensive language check. I understand that you may use "a native speaker", however, that does not guarantee the use of proper grammatical sentences and a good flow of the text for a reader.

Reviewer 3 Report

I am not sure if the authors have adequately/accurately addressed the study aims according to the study title. The aim of the study does not seem to be “how to prevent sepsis” (as suggested by the title) .

In data collection, I think the convenience sample should not be adopted, because this research seems to focus on a local issue of Croatia.

What exactly is the contribution of conducting an assessment of the educational content that students receive in a particular year of study?

The sampling is unequal/unbalanced from the point of “Year of study” -First year (65.4%), Second year (27.3%), and Third year (7.3%). Certainly, the results of this research require confirmation in a wider sample, for generalisability purposes. 

However, the authors have ignored these key opinions.